# The sound of reading: Color-to-timbre substitution boosts reading performance via OVAL, a novel auditory orthography optimized for visual-to-auditory mapping

Roni Arbel[1,2]*, Benedetta Heimler[1,3], Amir Amedi[1,3]

**1** Department of Medical Neurobiology, Hebrew University of Jerusalem, Hadassah Ein-Carem, Jerusalem, Israel, **2** Faculty of Medicine, Hebrew University of Jerusalem, Jerusalem, Israel, **3** The Baruch Ivcher Institute For Brain, Cognition & Technology, The Baruch Ivcher School of Psychology, Interdisciplinary Center Herzliya, Herzliya, Israel

\* roni.arbel@mail.huji.ac.il

**Data Availability Statement:** The data is located in OSF: (https://osf.io/fu5tg/).

**Funding:** AA is supported by a James S. McDonnel Foundation scholar award (no. 652 220020284)

## Abstract

Reading is a unique human cognitive skill and its acquisition was proven to extensively affect both brain organization and neuroanatomy. Differently from western sighted individuals, literacy rates via tactile reading systems, such as Braille, are declining, thus imposing an alarming threat to literacy among non-visual readers. This decline is due to many reasons including the length of training needed to master Braille, which must also include extensive tactile sensitivity exercises, the lack of proper Braille instruction and the high costs of Braille devices. The far-reaching consequences of low literacy rates, raise the need to develop alternative, cheap and easy-to-master non-visual reading systems. To this aim, we developed OVAL, a new auditory orthography based on a visual-to-auditory sensory-substitution algorithm. Here we present its efficacy for successful words-reading, and investigation of the extent to which redundant features defining characters (i.e., adding specific colors to letters conveyed into audition via different musical instruments) facilitate or impede auditory reading outcomes. Thus, we tested two groups of blindfolded sighted participants who were either exposed to a monochromatic or to a color version of OVAL. First, we showed that even before training, all participants were able to discriminate between 11 OVAL characters significantly more than chance level. Following 6 hours of specific OVAL training, participants were able to identify all the learned characters, differentiate them from untrained letters, and read short words/pseudo-words of up to 5 characters. The Color group outperformed the Monochromatic group in all tasks, suggesting that redundant characters' features are beneficial for auditory reading. Overall, these results suggest that OVAL is a promising auditory-reading tool that can be used by blind individuals, by people with reading deficits as well as for the investigation of reading specific processing dissociated from the visual modality.

and by a European Research Council Consolidator-Grant (no. 773121). The funders had no role in study design, data collection and analysis, decision to publish, or preparation of the manuscript.

**Competing interests:** The authors have declared that no competing interests exist.

# Introduction

"Once you learn to read, you will be forever free."

- Frederick Douglass

Among the most important cognitive skills, reading not only enables communication and information acquisition, but exerts a wide impact on brain organization and neuroanatomy as revealed when comparing literates to illiterates [1–6]. This includes the further refinement of brain mechanisms tailored to spoken language processing and phonemic awareness, and the development of anatomical structures and language-oriented functional-selectivity across the visual and lingual cortices, alongside a series of behavioral advantages in various domains [3, 6]. Readers, for instance, showed better low-level sensory abilities compared to non-readers [7–12] improved phonological processes and speech processing [11–16] and improved verbal memory [17–19]. Some evidence even hint to the possibility that reading influences higher-level cognition such as problem solving or planning, although the effect of schooling is difficult to filter out in these cases [20] (see for reviews [1, 21]).

While nowadays in developed countries literacy rates among sighted individuals are high and constantly increasing, literacy rates among blind individuals who rely on tactile-based systems such as Braille is only about 10% and declining [22]. In addition to possible effects on neuroanatomy and various related behaviors, this low literacy rate is associated with lower academic achievement, lower employment status and even lower general satisfaction with life [23, 24]. So, why is Braille literacy in decline? The first problem is related to the accessibility of Braille code. In contrast to print based systems that benefit from thorough instruction in schools, and are easily accessible by sighted readers both in physical form (i.e., books) and through all types of electronic devices, Braille code often lacks access to dedicated instruction, thus preventing many blind children to properly learn it [25]. Moreover, Braille texts require either physical production of heavy and expensive books, or expansive electronic Braille displays that require specialized hardware [25]. The second, and even more crucial issue, concerns the difficulties in mastering a tactile script, which requires enormous effort on the sharpening of tactile sensitivity. The distance between dots in a Braille cell is close to the tactile acuity limit [26], which makes tactile reading acquisition a long and difficult process especially for late blind individuals who typically have lower tactile sensitivity [27–29]. Indeed, several pieces of evidence demonstrate that the onset of blindness largely determines proficiency in Braille reading in adulthood: congenitally and early blind individuals are able to read 80–120 Braille words-per-minute and more [30–32], whereas late-blind people usually read two to three times slower and many of them do not succeed to properly learn Braille at all [27, 31, 33]. The decline of Braille reading is accompanied by an increased use of audio technologies such as audiobooks and other text-to-speech conversion algorithms used in tandem with digital devices. The lack of training required for the use of these programs, their low-cost and easy installation and the fact they don't require dedicated hardware make them appealing for blind and visually impaired individuals, especially students and others who are required to read large pieces of information, although initial evidence suggests that Braille reading is still the preferred medium by students [34]. This reported preference for Braille-reading appears to be in line with the predictions coming from the seminal and influential theory of the Simple View of Reading positing that reading comprehension is determined by two cognitive capacities: decoding words through alphabetic coding, and language understanding [35, 36]; see [37] for more recent empirical evidence). Under this framework, audio listening technologies, such as audiobooks, are not classified as reading as they lack the decoding aspect and only rely on

language understanding. Although initial evidence suggests that semantic comprehension is not worse for audiobooks than print-reading [38–40], other evidence highlights various advantages of print-reading over passive listening including extensive changes triggered by literacy both in white and grey matter within the language-system, with reading activating a broader brain network compared to listening [1, 41]. Research in this field, however, is still inconclusive and more work is needed to fully characterize the differences and similarities between these two processes.

Taken together, all the aforementioned evidence clearly suggests that typical reading acquisition exerts many benefits for neural and behavioral efficiency, thus highlighting the great need to create additional and alternative full reading options for the blind and visually-impaired population which in turn rely not only on language understanding (as audio listening technologies), but also on active word decoding (as print-reading systems). Such novel systems should be cheap, easily accessible, as well as both relatively easy and quick to master also in adulthood, in contrast to current tactile-based systems [42]. We propose that creating a reading system that is based on discrimination of auditory patterns, rather than tactile ones as Braille, could be a promising solution. Indeed, it is known that the auditory system has a higher resolution in both time and frequency than that of the somatosensory receptors on the skin [43], thus potentially substantially reducing training time while simultaneously also potentially increasing overall reading speed. Previous attempts in this direction, aimed at transforming visual letters into auditory spatio-temporal patterns using visual-to-auditory Sensory-Substitution Devices (SSDs), namely devices that transform visual information into audition using algorithms that preserve exact shape and location of objects [44, 45]. These studies demonstrated that it is indeed possible to read via audition, ultimately showing that auditory reading activates the same region in the ventral visual stream, the Visual Word Form Area (VWFA), as visual and Braille reading [46–49]. However, this type of SSD reading did not exit the lab's walls, and was not adopted as a reading tool by the blind community. One possible explanation for this outcome, is that visual letters' orthography may not be optimal for conversion to the auditory modality, ultimately lengthening the process of reading, hampering its fluidity and potentially even increasing the necessary training time. Indeed, similar conclusions were reached in the creation process leading to tactile-reading systems. Initial attempts in this direction aimed at reading via embossed visual letters, which resulted in a cumbersome process [50]. Tactile reading accuracy was significantly improved following the development of tailored tactile-based systems such as the Braille code [51]. We therefore hypothesize that instead of a direct translation of the visual orthography into audition, auditory reading may be facilitated by using a script designed specifically for this sensory modality.

Therefore, we created an auditory-based alphabet called OVAL, which characters are composed by combining together features from Braille systems (i.e., in Braille each character is characterized by a unique spatial pattern) and Morse code (i.e., in Morse code, each character is characterized by a unique temporal pattern of "dots" and "lines") (Fig 1A). OVAL characters were first created in their visual form, and then sonified by the visual-to-auditory EyeMusic SSD, creating unique auditory spatio-temporal patterns that we term "audemes", auditory transformed graphemes (Fig 1B).

In addition, seminal findings in tactile shape analyses demonstrated that the addition of texture variations to tactile shapes, namely a redundant, discriminative feature, enhanced shape discrimination performances [52]. It was thus suggested that tactile reading might also benefit from the addition of redundant discrimination information. However, because the systematic addition of texture to Braille is not feasible due to high costs and the fact that Braille characters wear out with use, the influence of redundant features on non-visual reading performance was never systematically tested. Research on proficient visual readers, demonstrated the opposite

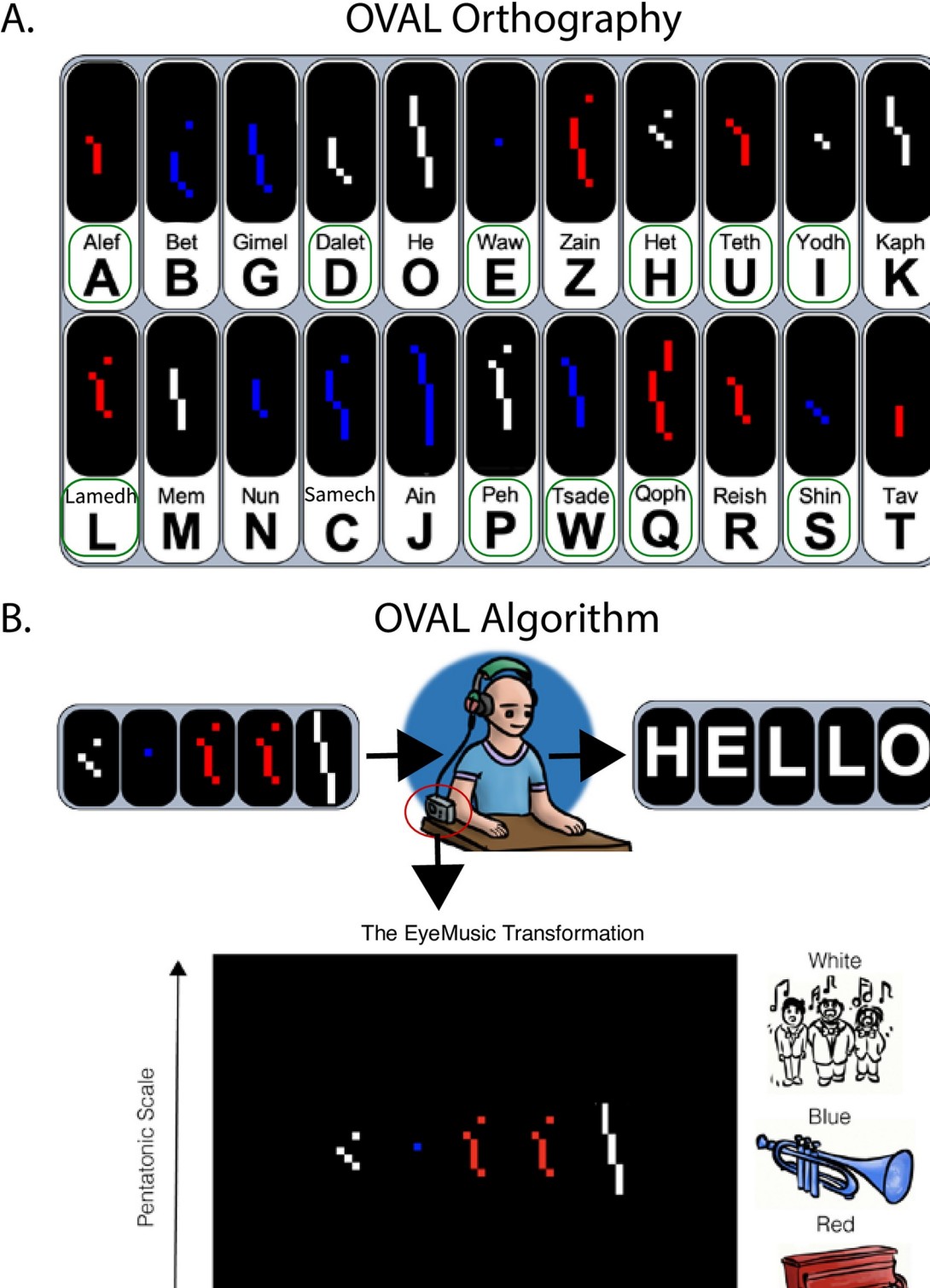

**Fig 1.** A. OVAL orthography. Visual representation of the Hebrew alphabet translated into the OVAL orthography. Trained letters are highlighted by a green square around the letter's Hebrew name and its correspondent transcription homologue in the Latin alphabet. Note that we depict here the Color OVAL. In the Monochromatic OVAL, all letters are white. B. OVAL Algorithm. OVAL visual letters were translated into sounds following the EyeMusic transformation algorithm: each image is scanned from left-to-right using a sweep-line approach so that x-axis is mapped to time (i.e., characters positioned more on the left of the image are heard first). Y-axis is mapped to the frequency domain using the pentatonic scale such that parts of a character which appear higher in the image, will be sonified with a higher pitch. Color is mapped to musical instruments. In the Color OVAL we used three colors: white, blue and red transformed into choir, trumpet and piano respectively. In this example we wrote with the OVAL the word "Hello".

outcome, namely that when adding redundant discriminative features to each letter, such as color, reading performance was hindered [53]. In contrast to Braille code, the addition of redundant discriminative features is easily implemented in the OVAL orthography. Indeed, the EyeMusic algorithm, which we use to create OVAL audemes, has the option to convert color into audition via timbre manipulations (i.e., different musical instruments) [44]. This in turn enabled us to directly investigate the influence of redundant discrimination information on auditory reading. One hypothesis was that reading performance will be enhanced when adding redundant features to OVAL characters, due to possibly easier discrimination between letters, as suggested in the tactile modality [52]. The alternative hypothesis was that such redundant information would hinder reading as shown in the visual modality [53].

The first aim of this work was to investigate the feasibility of OVAL as a quick-to-learn and efficient reading system. Thus, we tested a group of blindfolded literate sighted participants on several reading tasks after 6-hours of a specific OVAL training. In addition, we aimed to investigate whether redundant features, such as color, facilitate or hamper auditory-reading. Therefore, we tested an additional group of blindfolded sighted participants using the exact same training and reading tasks, though using this time a color (multi-instrument) version of OVAL. For this first investigation into auditory reading we tested sighted rather than blind participants to assess whether our auditory orthography is a valid non-visual reading system. This allowed us to exclude possible confounds in our results such as the level of literacy (all our participants acquired reading at typical developmental stages) or auditory training and experience (which is known to be enhanced in the blind population [54–57]). These results can be used as a baseline for quick OVAL learning, which could be then used in future studies as comparison to results acquired in other populations such as blind and visually impaired, as well as synesthetes or people with reading difficulties, among others.

## Material and methods

### Participants

A total of 18 sighted individuals took part in the experiments (age: 22.67 years SD; 4.54, 7 males). All participants were right handed, literate, native Hebrew speakers without any diagnosed neurological conditions (including dyslexia or other learning disabilities) and all had normal or corrected-to-normal visual and hearing abilities. In addition, all participants were naïve to the EyeMusic visual-to-auditory SSD as well as to any other SSDs. The experiment was approved by the ethical committee of the Hebrew University of Jerusalem. All participants signed a consent form before starting the experiment and received monetary compensation for their participation. Participants were assigned to one of two groups (9 participants in each), the "Color" group or the "Monochromatic" group and underwent OVAL training and experiments as described below. Participants were blindfolded throughout both training and experimental procedures.

## OVAL system

The OVAL auditory script is a novel alphabetic orthography designed for optimal compatibility with the EyeMusic visual-to-auditory sensory substitution device (SSD). Each letter of the alphabet is created in its visual form using a unique combination of vertical lines and dots, which vary in number as well as in their spatial layout, and optional color (depending on the training group, see below) (Fig 1A). OVAL letters are transformed to sound following the three principles of the Eye-Music SSD: 1. Y-axis location: the higher on the screen a feature of a character is positioned, the higher in pitch it will be sonified, and the lower on the screen a feature of a character is positioned, the lower in pitch it will be sonified, using a pentatonic scale; 2. X-axis location: each image is scanned from left to right in a column-by-column manner, so that users will hear first the feature of a character positioned more to the left. 3. Color: Colors are conveyed via timbre manipulations (i.e., different musical instruments). The EyeMusic algorithm can currently convey five colors (white, blue, yellow, red, green) and silence is conveyed with black. Here we used 3 colors: white (choir), red (Reggae organ), and blue (brass instruments) (Fig 1B). In the Monochromatic group, all letters were white, while in the Color group each letter had a given timbre assigned to it, thus color served as another feature helping to differentiate among the letters (Fig 1B).

More specifically, OVAL characters are composed by combining together features from Braille systems (i.e., in Braille each character is characterized by a unique spatial pattern) and Morse code (i.e., in Morse code, each character is characterized by a unique temporal pattern of "dots" and "lines"). Letter orthography was based on the combination of dots and lines as in Morse code. However, differently from Morse characters' configurations, in OVAL lines were vertical rather than horizontal. This was done to take advantage of pitch variations characterizing the Eye-Music algorithm (i.e., see y-axis transformation in the paragraph above; Fig 1B) allowing the creation of a unique spatial configuration for each character, a key feature of all orthographies. This choice also allowed each character to be played more quickly than if horizontal lines were used (i.e., see x-axis transformation in the paragraph above; Fig 1B). Morse characters are composed of up to 4 elements. In the current tested version of the OVAL algorithm, each element is played for 100 ms. However, to further limit the temporal length of letters and thus permit faster reading, in case a letter was composed of 4 components, the 4th component was added on top of the third, thus creating a "dual component" (an EyeMusic column containing two features of a character played simultaneously). Therefore, the maximum length of OVAL characters was 300 ms (3 x-axis EyeMusic columns). Similarly to Morse code, more frequent letters use shorter combinations, thus the most frequent letters in the Hebrew orthographic system were represented by short spatio-temporal configurations (i.e., 1 EyeMusic column: 100 ms), a bit less frequent letters by longer spatio-temporal configurations (i.e., 2 EyeMusic columns: 200 ms), and least frequent letters by even longer spatio-temporal configurations (i.e., 3 EyeMusic columns: 300 ms). Finally, in both training and experimental procedures, independently of the length of OVAL characters/words, each EyeMusic scan of an OVAL display lasted 4 seconds.

In this experiment we trained participants on identifying 11 OVAL characters, half of the Hebrew alphabet (see Fig 1A; for details see section below on OVAL training). These chosen letters were controlled for frequency in the language. Specifically, the letters were arranged in order of frequency in the Hebrew language. Every other letter was chosen as part of this experiment, thus achieving a set of 11 letters with varying occurrence in the language.

## Experimental apparatus

All participants sat in-front of a laptop computer and wore a set of headphones through which they received EyeMusic information throughout the training and experimental procedures (Fig 1B).

## Experimental procedure

Before OVAL training, participants performed an OVAL character discrimination task which served as a baseline test for discrimination of OVAL characters. Then participants went through 6-hour specific OVAL training which consisted of 3 training sessions, lasting 2 hours each. During each two-hour sessions, participants took short breaks when they felt tired to prevent excessive fatigue and to allow the maintenance of focus during OVAL learning. Following training, participants repeated the OVAL character discrimination task, along with three additional experiments, a task discriminating trained from untrained letters, a letter identification task on the trained characters, and a reading task using both words and pseudo-words of up to 5 characters comprised by any of the trained OVAL letters. The training sessions and the after-training experimental session needed to be completed within six days with the experimental session being held the day after the last training session took place. Training protocols and experiments were exactly the same between the two groups (Monochromatic and Color groups). The presence or absence of colored OVAL letters was the only feature differing between groups. Below, we will describe in details all aspects of the experimental procedure. A power analysis using G*power [58] showed that to achieve a power > 80% with a priori alpha set at 0.05 to study within-between variable interactions with partial eta squared of 0.14 in the ANOVAs, 5 participants are needed in each group.

All experiments were performed using Presentation® software (Version 18.0, Neurobehavioral Systems, Inc., Berkeley, CA, www.neurobs.com)

## OVAL training

The OVAL training protocol was designed to teach participants the identification of 11 Hebrew OVAL letters, using the EyeMusic SSD and read short words and pseudo-words consisting of those letters formed by up to 5 characters.

Following baseline "letter discrimination" task, it was explained to the participants that the sounds they just heard conveyed letters in a novel auditory orthography. Then, the basic algorithm of the EyeMusic was explained briefly (i.e., the rules of the algorithms regarding the x- and y-axes transformations, and the color transformation for the color training group). The training on reading with OVAL unfolded as follows: First, the letter "waw", the most common letter in the Hebrew orthographic system, was played. This is represented by a single dot in the OVAL orthography, translated into audition, according to the EyeMusic algorithm, as a brief, single tone -this letter was blue for the Color group, i.e., played via the brass instrument. Participants were then encouraged to decipher such pattern independently, based on the algorithm of the EyeMusic that was briefly explained to them. Most participants could report that the sound they heard represented a dot. The experimenter then explained the related color (timbre) information (for the Color group), and provided them with the phonological meaning of the audeme. The next letter, "alef" was then played. Again, participants tried to decipher the pattern based on the algorithm transformation rules with feedback from the experimenter if needed. Then for the Color group, the color/timbre information was disclosed and finally the phonological meaning of the audeme was provided. Two more letters were introduced, and after a short repetition of the 4 learnt OVAL letters, participants were instructed to read short words composed of these characters. When participants became able to read these words, 3 more letters were introduced following the same procedure described above, then participants were asked to read words formed by any combination of the 7 letters they learned. Then 2 more OVAL letters were introduced, and then participants read words composed by any of the 9 letters learned; then the 2 last letters were added and participants read words composed by any of the 11 learned letters. The final part of the training consisted of reading words

and pseudo-words containing all 11 learned letters. The majority of participants learned 7–9 OVAL letters (and to read short words composed by them) within the first 2-hours training session, with some participants, from the "Color" group even succeeding to learn all 11 OVAL characters within this first training session. When necessary, the remaining 2–4 letters were learned in the second training session, and the rest of the training program was dedicated to word/pseudo-words reading and 2-words combinations reading. The order of introduced letters, as well as the order of presented words/pseudo-words was exactly the same for all participants in both training groups, with little variation to address specific individual needs. By the end of the training program, all participants were able to correctly read the list of words/ pseudo-words presented during training. Importantly, the words/pseudo-words used in training differed from those used in the experiment.

## Behavioral experiments

**Experiment 1 –letter discrimination.**   The ability to discriminate different OVAL characters was tested both before OVAL training ("pre-training") and after training ("post-training"). In the pre-training stage, all subjects were naïve to the purpose of the study, to the OVAL orthography and to the EyeMusic SSD algorithm.

To test the baseline character-discriminability of the OVAL auditory orthography as well as training induced improvement in discrimination, similarly to what has been done in Braille, we created a letter-discrimination task with forced-choice responses [59, 60]. Each trial consisted of a pair of OVAL characters which could be either identical or different. The task of participants was to decide whether the characters played were same or different by pressing one of two possible response keys with the index and ring fingers (response keys for same/different were counterbalanced among participants). Each trial began with a silence period of 1500–1800 ms and then a pair of OVAL characters was played consecutively with a 700 ms pause between the two presented letters until participants' response. Each pair of characters was presented within 4 seconds intervals (i.e., the time used by the EyeMusic to scan each OVAL display).

The experiment comprised 242 trials in total divided in two separate blocks of trials, with a break of up to 90 seconds between blocks. The 242 trials consisted of all possible combinations of the 11 experimental OVAL letters, each repeated twice in a random order with the constraint of avoiding two identical trials one after the other. Correct rates were assessed.

After training, participants repeated the same task using a different sequence of trials presentation, which was created based on the same criteria of randomization of the pre-training sequence.

Both experiments lasted ~20 minutes each.

**Experiment 2 –discrimination of trained vs. untrained letters.**   In order to further investigate how well the OVAL characters could be discriminated between each other, and to exclude that OVAL characters selected for this experiment were by chance easier to discriminate, we presented to participants, after training, a discrimination task among all OVAL letters (all 22 letters forming the Hebrew orthographic system; Fig 1A). Participants were asked to decide if a letter belonged to the cohort of trained letters (i.e., familiar letters), or not (i.e., untrained letters), and provide their response by pressing one of two possible response keys with the index and ring fingers (response keys for trained/untrained letters were counterbalanced among participants). Each trial began with a silence period of 1500–1800 ms and then one of the 22 OVAL letters was presented repeatedly within four seconds intervals.

During the whole experiment, each OVAL letter was repeated 3 times for a total of 66 trials presented in a random order with the constraints that the same letter could not be repeated

more than twice consecutively, and that repetitions of the same letter were not all appearing in the same chunk of the experiment. There were two possible trials sequences to control for order biases, which were presented to participants in counterbalanced order. The number of correctly identified letters (correct rate–CR) and reaction times were collected. The whole experiment lasted ~7 minutes.

**Experiment 3—letter identification of trained characters.** In order to test basic letter identification ability following training, participants were asked to name each of the 11 experimental letters they received training on. Each trial began with a silence period of 1500–1800 ms and then one OVAL letter was presented repeatedly within four seconds intervals. Participants were instructed to press the space bar as soon as they identified the letter and then to give their response orally (which was inserted in the computer by the experimenter). Then they pressed again the space bar to move to the next trial. During the experiment, each letter was repeated 6 times for a total of 66 trials in a random order with the constraints that the same letter could not be repeated more than twice consecutively, and that repetitions of the same letter were not all appearing in the same chunk of the experiment. There were two possible trials sequences to control for order biases, which were presented to participants in counterbalanced order. The number of correctly identified letters (correct rate–CR) and reaction times were collected. The experiment lasted ~10 minutes.

**Experiment 4 –words/pseudo-words reading task.** To test participants' reading proficiency, after training, they underwent a words/pseudo-words reading task. 68 stimuli, half words and half pseudo-words with various length, all of which had not been introduced during training, were presented in a random order, with the constrains that not all same-length words appeared in the same chunk of the experiment. Pseudo-words were created by switching one letter of an experimental word, with a different letter from the pool of trained letters. Each word/pseudo-word consisted of two to five letters (20 short words and 20 pseudo-words of either 2 or 3 letters; 14 long words and 14 long pseudo-words of either 4 or 5 letters, using all the 11 characters learned during training). Each stimulus was played repeatedly until participants' response. Participants were informed at the beginning of the task that stimuli could be either words or pseudo-words and were instructed to listen to the whole stimulus and press the space bar as soon as they completed the reading of each word/pseudo-word. Then they were asked to provide their response orally, namely saying the word/pseudo-word aloud (and the experimenter entered it in the computer) and pressed the space bar to move to the next trial. Spelling of Hebrew words and pseudo words correspond most of the time to their pronunciation. In the few cases in which spelling was not directly obvious from pronunciation, we asked participants to spell the word aloud. Each trial started with a tone marking its beginning lasting ~20 ms. During the experiment, each word/pseudo-word was presented within 4 seconds intervals. The number of correctly read words/pseudo-words (CR) and reaction times were recorded. The length of the experiment varied based on participants' performances, ranging between 20–40 minutes.

## Data analysis

All statistical analysis was conducted using JASP software (version 0.11.1), using t-tests and mixed repeated measures ANOVA. Post-hoc tests were conducted using Bonferroni correction. Effect sizes are reported using partial eta squared in ANOVAs or using the Cohen's d representing pooled-SD in t-tests. Below, we present all the results comparing performances between the two groups (monochromatic and color OVAL readers), separately for each one of the experiments. Reaction times are calculated only for correct trials.

## Results

### Experiment 1—letter discrimination task

In the pre-training task, thus before learning the phonological meaning of the auditory spatio-temporal patterns, we aimed at testing basic auditory discrimination among the various OVAL letters. OVAL monochromatic readers' average success was 88.4% (± 9.84%—standard deviation (SD)) while OVAL color readers' average success was 98.87% (SD ± 0.92%). Two t-tests against chance (50%) confirmed that performances in both groups was above the chance level (Monochromatic readers: $t(8) = 11.70$, $p < .001$, $d = 3.9$; Color readers $t(8) = 158.55$, $p < .001$, $d = 52.85$), thus indicating that participants were able to discriminate among OVAL characters only based on their distinctive auditory properties.

When we repeated the same experiment after training, thus investigating the benefit of phonological training on OVAL letters discrimination, we observed that OVAL monochromatic readers' average success was 95.27% (SD ± 3.6%) while OVAL color readers' average success was 99.7% (SD ± 0.34%) (Fig 2A). To test whether there was any significant difference between the performances of the two groups and whether explicit reading instruction induced improvement in discriminability of letters, we performed a mixed repeated measures ANOVA using average success rates as the dependent variable, with "Group" (color vs. monochromatic) as between-group factor and "Learning stage" (pre-training vs. post-training) as within-group factor. This ANOVA revealed a main effect of Group ($[F(1,16) = 11.3$, $p = .004$, $\eta^2_p = 0.41]$), due to overall higher accuracy in the Color compared to the Monochromatic training group (Color readers: M = 99.17% SD = 0.85%; Monochromatic readers: M = 91.83% SD = 8.0%). Also the main effect of "learning stage" was significant ($[F(1,16) = 11.4$, $p = .004$, $\eta^2_p = 0.42]$), due to overall higher accuracy in the post- compared to the pre-training task in both groups (pre-training = 93.52% SD = 8.60%; post-training = 97.47% SD = 3.36%). Finally, we also observed a significant interaction effect between learning stage and group ($[F(1,16) = 6.28$, $p = .02$, $\eta^2_p = 0.28]$). Post-hoc analysis revealed that the difference in performance between the two groups was mainly due to the Monochromatic training group performing significantly worse before training compared to the Color group ($p = .002$), while performances between groups were not significantly different post-training ($p = .53$) (Fig 2A).

### Experiment 2 –discrimination of trained vs untrained OVAL letters

Considering participants were trained on half of the alphabet, we wanted to ensure that discrimination of trained letters generalized to the entire Hebrew Oval alphabet. Therefore, we included after training, a discrimination task of all characters (22 letters forming the Hebrew orthographic system; Fig 1A), in which participants had to decide if a given character was familiar or not.

Results showed that both groups could successfully differentiate trained letters from untrained letters, with monochromatic readers correctly identifying 85% (SD ± 0.10%) of the letters on average, significantly above chance level (50%) ($t(8) = 10.31$, $p < .001$, $d = 3.4$). Color readers identified on average 94% (SD ± 0.051%) of the letters, which resulted significantly above chance level as well ($t(8) = 26.2$, $p < .001$, $p = 8.7$). Color group performed significantly more accurately than monochromatic group ($t(16) = 2.56$, $p = .023$, $d = 1.181$) (Fig 2B).

Color readers were also significantly faster in such discrimination, taking on average 1.36 seconds (SD ± 0.77 sec) to respond comparted to Monochromatic readers who took 2.68 seconds on average (SD ± 0.02 sec; $t(16) = 3.104$, $p = .007$, $d = 1.463$) (Fig 2C). Despite this difference, both groups identified if a letter was trained or not within the first character presentation (each character was presented within 4 seconds intervals).

# Performance on single OVAL letters

## Letter discrimination

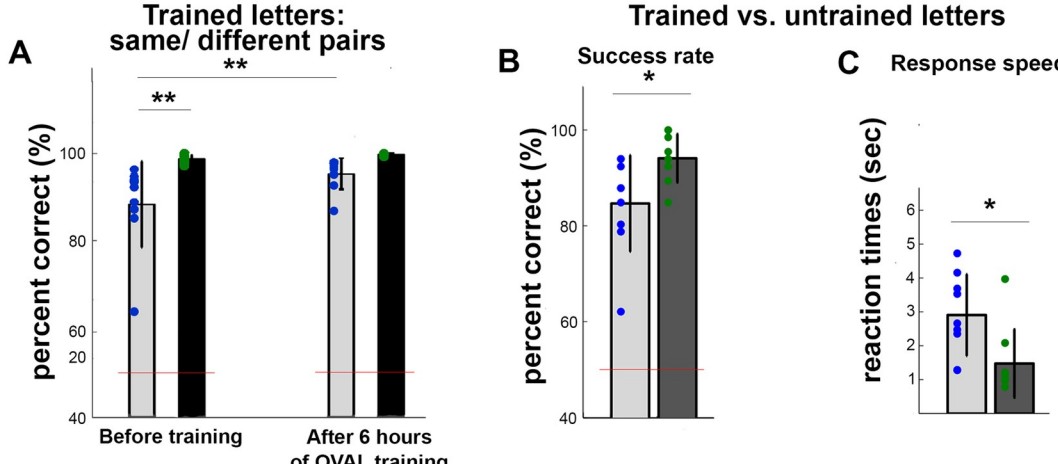

## Identification of trained characters

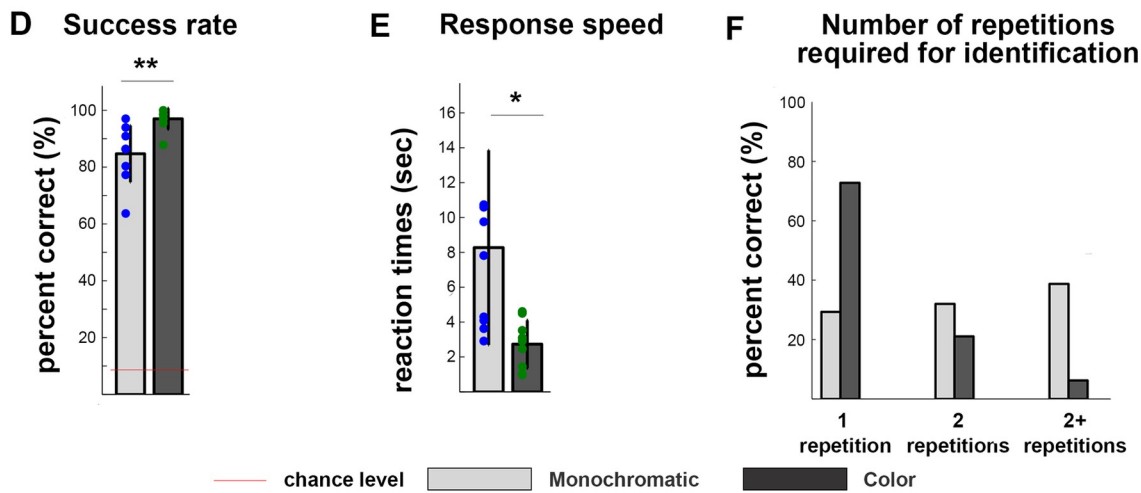

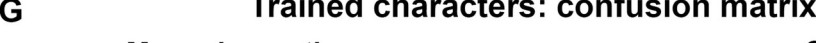

### G    Trained characters: confusion matrix

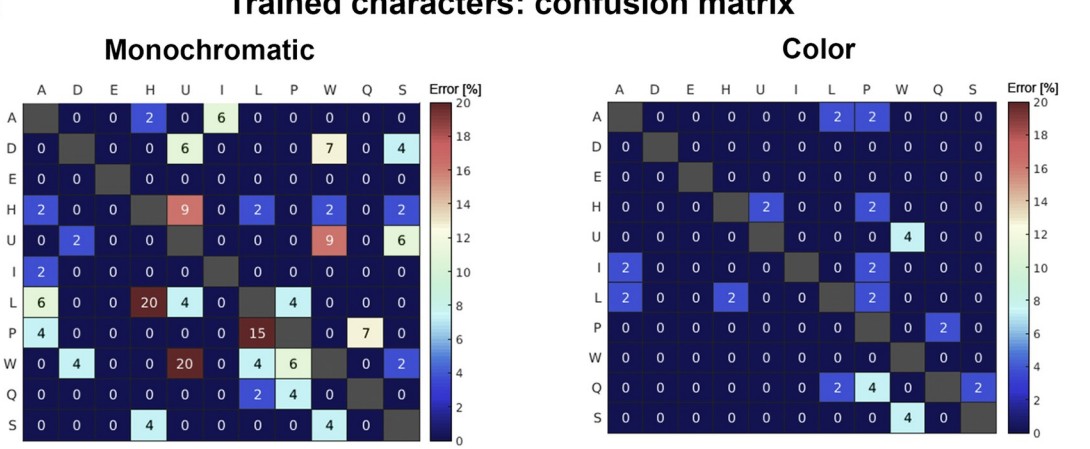

**Fig 2.** A. Comparison in the auditory discrimination among trained OVAL characters between pre- and post-training performances. Although pre-training discrimination was already high, indicating easy discriminability of the OVAL audemes, both groups significantly improved after-training, reaching ceiling effect. The Color group outperformed the Monochromatic group in the pre-training performance. B-C. After training, we tested the discriminability of trained vs untrained OVAL letters. B. Both groups reached high-success rate in this task, but Color readers significantly outperformed Monochromatic readers. C. Color readers could also identify trained from untrained letters significantly faster than Monochromatic readers. D-E. During the letter identification task, both groups show high accuracy, but the Color group achieved higher success rates (D). In addition the Color group also showed shorter reaction times (E). F. In the majority of trials, Color OVAL readers required only one presentation of a letter in order to correctly identify it, while Monochromatic readers require an additional repetition on average. In A-F error bars represent standard deviations (SD). Asterisks represent statistically significant differences: ** p<0.005; * p<0.05. G. Pattern of errors during identification of trained characters depicted in a confusion matrix. Each cell represents the percent of errors participants committed in identifying each letter pairs, reported separately for the Monochromatic (left) and Color (right) groups.

## Experiment 3 –letter identification task

Apart from basic sensory discrimination, we tested weather participants learned to successfully identify the 11 OVAL letters they learned during our relatively short training period. Results showed that both groups could successfully identify trained letters, with Monochromatic readers correctly identifying 84.7% (SD ± 10.0%) of the letters on average, significantly above chance level (9%) ($t(8) = 22.69$, $p < .001$, $d = 7.56$). Color readers identified 96.97% (SD = ±3.7%) of the letters on average which resulted significantly above chance level as well ($t(8) = 71.11$, $p < .001$, $d = 23.7$). In addition, the Color group performed significantly more accurately than the Monochromatic group ($t(16) = 3.46$, $p = 0.003$, $d = 1.63$) (Fig 2D). To investigate the error pattern of participants' responses and thus highlight whether participants had difficulties in identifying specific OVAL letters, we created a "confusion matrix" with percent of mistakes for each possible pair. Results showed no specific pattern of mistakes in the Color group which overall did very little identification errors (see Fig 2G). Results from monochromatic readers were more informative, showing they committed more mistakes in identifying Lamedh (L) as Het (H), and Tsade (W) as Teth (U), pairs of letters with relatively similar visual orthography (Figs 1A; 2G).

In addition to their superior accuracy Color readers were significantly faster than monochromatic readers in correctly identifying OVAL characters ($t(16) = 2.89$, $p = 0.01$, $d = 1.36$) (average Color readers = 2.73 sec, SD ± 1.37 sec; average Monochromatic readers 8.27 sec, SD ± 5.59 sec). These results show no evidence of speed-accuracy trade-off with the addition of color (Fig 2E). Indeed, while Monochromatic readers correctly identified the letters within about 2 presentations of each character, Color readers required only one presentation on average [Fig 2F].

## Experiment 4—word/pseudo-word reading task

Finally, we assessed whether the letter identification capacity of our participants following this short reading-specific training, extended to read new, untrained words and pseudo-words composed of the trained 11 OVAL letters. Experimental stimuli were not presented during training and were not limited to a specific list, thus chance level performance is close to zero. Monochromatic readers correctly read 61.6% of the words/pseudo-words on average (SD ± 25.0%) while Color group readers correctly read 85.7% of the words/pseudo-words on average (SD ± 5%) (Fig 3A).

We entered the average accuracy rate for each participant into a mixed repeated-measures ANOVA with Group (Monochromatic vs. Color) as between-group factor, stimulus-type (words; pseudo-words) and stimulus-length (short, i.e., 2–3 characters; long, i.e., 4–5 characters) as within-participants factors. This ANOVA revealed a significant main effect of Group ([$F(1,16) = 8.021$, $p = .012$, $\eta^2_p = 0.33$]) due to overall higher accuracy in the Color compared

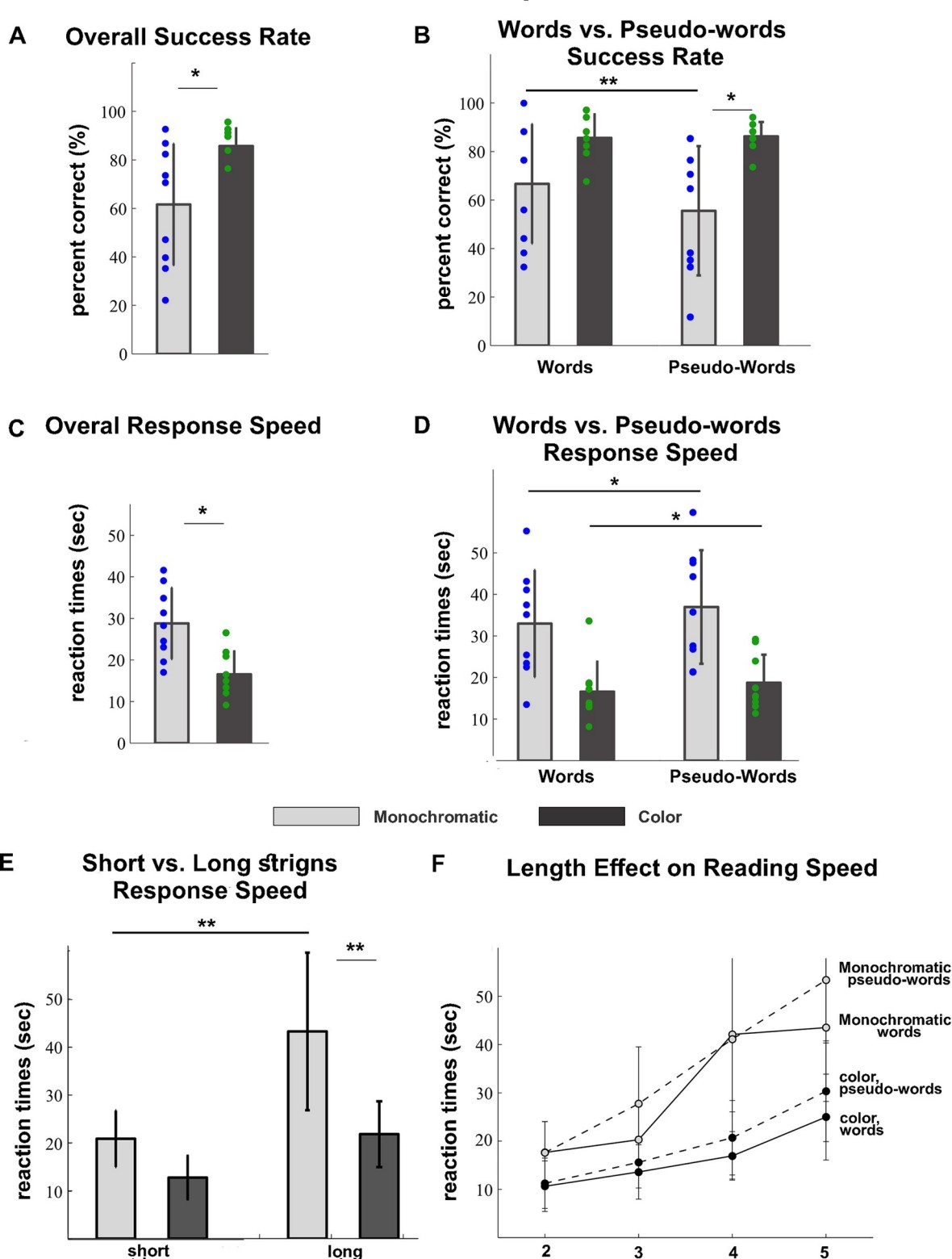

**Reading with OVAL after 6 hours of training:
Untrained words and pseudo-words**

**Fig 3. After only 6-hours of training both groups could successfully read OVAL strings.** A. Color readers were overall more accurate than Monochromatic readers. B. Interaction Group*Word-Type (words; pseudo-words): Both groups were more accurate to read words than pseudo-words but Color readers showed an advantage in reading pseudo-words compared to Monochromatic readers. C. Color readers were overall faster than Monochromatic readers in reading OVAL strings D. Both groups read words faster than pseudo-words. E. Interaction Group*Stimulus-Length. Both groups tended to read long strings quicker than short strings. However, Monochromatic readers read long words significantly slower than Color readers. Post-hoc analyses also revealed that Monochromatic readers were significantly slower when reading long strings compared to short ones, while Color readers, did not significantly differ in the reading speed of short and long strings. F. Length effect on reading speed. This is a complete representation of speed performances when reading OVAL strings of various length, plotted separately for each group, and for words and pseudo-words. Monochromatic readers drastically increased their reading speed when length of the OVAL strings increased. Color readers tended to show the same pattern but less prominent.

to the Monochromatic training group. In addition, also the main effect of stimulus-type was significant ([F(1,16) = 8.3, $p$ = .01, $\eta^2_p$ = 0.34) due to overall higher accuracy when reading words compared to pseudo-words in all participants (words: M = 76.14% SD = 20.60%; pseudo-words: M = 70.91% SD = 24.50%). Finally, also the interaction between Group and stimulus-type was significant ([F(1,16) = 10.96, $p$ = .004, $\eta^2_p$ = 0.41). Post-hoc analyses revealed that between-groups differences in performances resulted from readers in the Monochromatic training group being significantly less accurate than readers in the Color training group in reading pseudo-words (Color readers: M = 86.27% SD = 5.9%; Monochromatic readers: M = 55.56% SD = 26.65%; $p$ = .012) (Fig 3B). No other factors were significant (all F-values<2.02).

When analyzing reaction times, we observed that Monochromatic readers identified an OVAL string after 28.81 sec on average (about 7 word's repetitions; SD ± 8.56 sec), while Color readers identified OVAL strings in 16.53 sec on average (about 4 word's repetitions; SD ± 5.53 sec) (Fig 3C). We entered individual average reaction times into a mixed repeated-measures ANOVA, with Group (Monochromatic vs. Color) as between-group variable, stimulus-type (words; pseudo-words) and stimulus-length (short, i.e., 2–3 characters; long, i.e., 4–5 characters) as within-participants factors. This ANOVA revealed a main effect of Group ([F (1,16) = 13.34, $p$ = .002, $\eta^2_p$ = 0.46]), due to overall faster responses in the Color training group compared to the Monochromatic training group. Importantly these results confirm that the significantly more accurate reading outcome reported in the Color compared to the Monochromatic group was also accompanied by an advantage in reading speed. In addition, similarly to the ANOVA on accuracy, also the main effect of stimulus-type was significant ([F (1,16) = 9.74, $p$ = .007, $\eta^2_p$ = 0.38]) due to all participants being faster in reading words than pseudo-words (words: M = 21.38 sec SD = 9.20 sec; pseudo-words: M = 24.14 sec SD = 10.27 sec) (Fig 3D). The main effect of stimulus-length was also significant ([F(1,16) = 29.63, $p$ < .001, $\eta^2_p$ = 0.65)]) due to faster responses for shorter than longer OVAL strings (short: M = 16.81 sec SD = 6.60 sec; long: M = 32.60 sec SD = 16.50 sec). Importantly, also the interaction between stimulus-length and group was significant ([F (1,16) = 6.00, $p$ = .029, $\eta^2_p$ = 0.27)]) (Fig 3E and 3F). Post-hoc analysis revealed that Color readers were faster than Monochromatic readers in reading long OVAL strings (p < .001). Indeed, Monochromatic readers read long strings slower than short strings (short: M = 20.87 sec SD = 16.50 sec; long: M = 43.31 sec SD = 16.49 sec; $p$ < .001), while Color readers did not take significantly longer to read long than short strings (short: M = 12.75 sec SD = 4.54 sec; long: M = 21.83 sec SD = 6.88 sec; $p$ = .15). No other main effect or interactions were significant (all F-values <0.151).

## Discussion

In this work, we showed the efficacy of the OVAL, a novel auditory-based orthographic system, for non-visual reading. The OVAL comprises all letters of the Hebrew alphabet designed

with unique visual patterns of vertical lines and dots. Such visual letters are then transformed into audition using the algorithm of a visual-to-auditory SSD, the EyeMusic, which in turn creates "audemes"–unique auditory spatio-temporal patterns representing exact shapes and locations of letters. Our results show that even before undergoing tailored OVAL training, literate sighted blindfolded participants with no previous training on the EyeMusic or any other SSDs, were successful in discriminating OVAL characters. Then, participants underwent 6 hours of OVAL-specific training during which they were introduced to 11 OVAL letters, and were trained to read words/pseudo-words of various lengths composed of the trained characters. After training, participants showed a significant improvement in the same OVAL letter discrimination task and successfully discriminated trained from untrained OVAL characters. Additionally, they successfully identified all the OVAL trained characters as well as read untrained OVAL words and pseudo-words. These results suggest that the OVAL auditory-reading script can be successfully learned in adulthood, after a relatively short specific training. Future studies with prolonged training and bigger sample size can further assess the properties of OVAL reading, thus investigating for instance reading performances on the other untrained half of the OVAL alphabet and on the full alphabet, the reading speed achieved after significantly longer training, text comprehension properties, as well as additional possible convergent or divergent properties of OVAL compared to print reading, such as invariance to location and font variations [61]. Intriguingly, our findings suggest that adding redundant features to OVAL letters (i.e., adding specific colors to OVAL characters transformed into audition via timbre manipulations) is beneficial for the final learning outcomes. Specifically, we tested in the exact same paradigm, two different groups of participants: one who was only exposed to a monochromatic version of the OVAL and another who was only exposed to a colored version of the OVAL (i.e., different letters always appeared in a specific color and were played by a specific musical instrument according to the EyeMusic transformation algorithm; Fig 1A and 1B). Results showed that participants from the Color group, performed better than the participants from the Monochromatic group in all of our experiments. Importantly this enhancement was expressed via both faster reading rates and higher accuracy rates, thus with no "speed accuracy trade-off".

## Auditory OVAL compared with tactile Braille: Insights for reading and SSD training

The main method for reading in the blind and low-vision individuals who cannot read via the visual modality, is the Braille system based on tactile identification of letters. However while some people read Braille fluently and well, many people find it difficult and cumbersome, if not impossible, to master it. This is especially true for those faced with blindness in adulthood who may lack the tactile sensitivity needed to read Braille, such as, but not exclusively, when blindness is in co-morbidity with diabetes, today one of the leading causes of late on-set blindness, which leads to decreased tactile acuity [62]. Prior to the invention of Braille, blind individuals were taught to read using a system of embossed letters [63] that was slow to decipher. Thus, tactile reading had become popular only with the invention of the Braille code that aimed at creating a system which maximizes the properties of the tactile modality rather than a direct translation of the visual print to a tactile format. Braille letters are based on a tactile code developed by a French army captain designed specifically to allow reading in darkness. Then, Louis Braille, blind himself, simplified the original code by reducing the number of dots conveying each letter, from 12 to 6 in order to fit a 2X3 grid that can be read with a stroke of a finger; this change considerably improved the speed and efficiency of tactile reading, establishing the Braille code known today [63, 64]. Braille reading peaked in the 1960's with 50% Braille

literacy rate among the blind but decreased dramatically in the last few decades. Today it is estimated that only 5% to 12% of the blind population currently uses it as the main method for information acquisition, and numbers are estimated to be even lower when considering only the blind adult population [22]. This drop in the use of Braille is due to a combination of reasons among which is the lack of proper Braille instruction, high-cost of Braille electronic devices, the demand for very high tactile acuity (but see [42], and inherent prolonged training times to become a proficient reader. This drop in Braille literacy was flanked by an increase in the use of text-to-voice converters or audiobooks. However, in contrast to these approaches that make information accessible only through listening, Braille reading and also the novel OVAL orthography, are active reading systems that require the users to apply audeme-phoneme correspondence and understand the rules of spelling, similarly to print reading systems [35, 36]; see [37] for more recent empirical evidence), which were shown to exert various brain and behavioral advantages compared to auditory-only methods [1, 41].

To our knowledge OVAL is the first non-visual orthography that provides an easy-to-learn reading approach that can also be easily acquired in adulthood. We propose that the successful OVAL outcomes are at least partially due to the fact that this novel orthography is built on auditory discrimination of unique spatio-temporal sound patterns which are discriminable among each other even before specific training on this orthography (Fig 2A). Such discriminability was especially high in the Color group, thus suggesting that even without any phonological association to the auditory patterns, basic auditory discrimination of the soundscapes benefited by the addition of redundant information, in this case timbre manipulations (see also [65] for another color-related advantage in soundscapes discriminations). One explanation for these results, might be that learning OVAL, differently from learning Braille, does not require to stretch the limits of human auditory perception, again particularly in the color group, ultimately both reducing training time and increasing its efficacy: Indeed it is known that the auditory system has higher resolution in both time and space compared to the receptors on the skin (43). Several previous studies conducted on blindfolded sighted Braille learners, provide convergent results supporting these conclusions: For instance, in one study blindfolded sighted participants were able to discriminate between same and different Braille letter pairs, with error rates dropping from approximately 37% pre-training to 25% post-training, which included 5 days of extensive tactile stimulation in addition to 20 hours of formal Braille training [59]. In another study, blindfolded sighted children, before training, were able to perform a Braille discrimination task on 4 Braille characters with error rates of little less than 11%. After training, a final tactile discrimination test on these trained characters showed accuracy rates of ~90% [66]. In addition, a recent study showed that sighted individuals performing a Braille discrimination task on the whole Braille alphabet were able to discriminate 37.5% of the letters prior to Braille training, and reached 91.85% following more than 10 hours of training specifically focused only on characters discrimination [60]. These studies highlight that not only pre-training discrimination rates with Braille were substantially lower than those obtained via the OVAL auditory orthography, but crucially, even post-training discrimination rates were still lower than those obtained via OVAL, despite longer Braille training time. Note however, that a recent work showed that tactile acuity thresholds were not directly related to Braille learning outcomes in blindfolded sighted adults [42]. This latter result suggests, in turn, that the impressive higher discriminability of OVAL compared to Braille characters may not be entirely due to the lower demands of the OVAL orthography on auditory sensitivity compared to Braille demands on the tactile one. An additional and not mutual exclusive explanation for the more efficient learning of OVAL compared to Braille reading is that the several dimensions of variability among OVAL characters facilitated their discrimination. While Braille characters differ from each other only in the presence or absence of a dot in one of six

possible locations [52], OVAL characters differ in shape, frequencies and duration (as well as color in the Color group). The fact that discrimination abilities of the Color group exceeded those of the Monochromatic group supports this possibility, and suggests that the more different characters are easily distinguishable among each other, the more learning is facilitated. Note that the inclusion of all possible letter pairs in the current design imposes an imbalance of the task, as more trials were "different" than "same". Future studies that include only a partial sample of letter-pairs that enables a balance between same- and different-trials can increase sensitivity of the current result.

Finally, in-line with our predictions, OVAL reading outcomes were more successful than reading outcomes after conversion of visual letters to SSD soundscapes [47, 49]. Specifically, we achieved with OVAL, higher letters' discrimination abilities and related reading performances in shorter training time compared to SSD presented visual letters [47, 49]. These outcomes are particularly significant if one considers that differently from letters-to-SSD readers, OVAL participants were naïve both to the concept of SSD, and to the letters' shapes prior to training, and received no tactile feedback on such shapes during training to facilitate their learning process [47, 49]. Additionally, OVAL offers faster reading rates than SSD presented visual letters, as OVAL characters are deliberately narrower than print letters, and thus are translated to inherently shorter soundscapes. Finally, unlike OVAL character discrimination that was high even before training and reached ceiling after training (Fig 2A), soundscapes of visual letters are not necessarily easy to discriminate, as some letters have very similar visual shapes, making them harder to disentangle via SSDs. Taken together, these results suggest that similarly to what have been demonstrated for the Braille code [63], a reading system specifically tailored to the auditory modality leads to better reading outcomes than a simple transformation of visual letters into audition.

Future studies will need to assess whether other forms of auditory reading might elicit similar outcomes than what we reported for OVAL, such as hearing each letter's name and creating a word from that sound (i.e., dictated spelling). Our prediction is that dictated spelling will elicit a set of behaviors and neural activations that will lie in between the outcomes of audiobooks listening and the outcomes of reading. Dictated spelling, differently from audiobooks, requires both decoding words through alphabetic coding and language understanding, namely the two cognitive capacities determining reading comprehension according to the seminal theory of the Simple View of Reading [35, 36]; see [37] for more recent empirical evidence). However, while in OVAL reading each character has a unique spatio-temporal patterns, which we believe facilitates perception and speeds comprehension, reading via dictated spelling relies strongly on visual imagery of the print alphabet of reference (which might not be similarly available to all users), in turn potentially making retrieving words more effortful. Note that dictated spelling might nonetheless prove to be a promising supportive training solution to teach the OVAL alphabet, or other non-visual orthographies, mediating for instance the passage from the print visual alphabet to non-visual orthographies.

Taken together, these encouraging results obtained with OVAL highlight also the benefits that OVAL training can exert on other skills both within and outside the SSD realm. First, because our participants were entirely naïve to SSDs before starting the experiment, the current results promote OVAL as a promising way to teach the principles of SSDs to those interested in mastering their use, such as people who are blind and visually impaired. Indeed, the introduction of color and shapes through the teaching of reading a script can potentially be more engaging and enjoyable for users than the use of simple lines and geometrical shapes in the initial steps of SSD training as typically carried out [44]. This may in turn motivate participants to persevere with SSD training. Furthermore, learning OVAL teaches to discriminate different notes on the same Y axis column (i.e., through learning to differentiate characters

with overlapping components, namely two characters' elements appearing on the same column such as in "P" and "H" -see Fig 1A), a task that has been shown to be challenging for SSD users [67]. Our results show that even participants from the Monochromatic group, who had no redundant cues to differentiate letters among each other, could successfully differentiate letters with overlapping features in the Y axis (Fig 2G), thus supporting the notion that OVAL training facilitates tuning of auditory discrimination skills. Another, non-mutual exclusive possibility is that OVAL letters are perceived as an auditory unified spatial patterns, rather than separate components on different locations on the Y axis, which result in successful Y-axis discrimination of the character's features. This in turn may suggest that OVAL learning might train holistic auditory processing and thus facilitate learning of more advanced SSD soundscapes that require the analysis and perception of multiple overlapping components such as soundscapes of human faces. Finally, because OVAL is based on Morse code, learning OVAL might also facilitate Morse learning, as letters in both systems use the same combination of dots and lines–although some nuanced differences in the spatial arrangements of the characters' components between the two systems are present.

## The boosting power of color information carried by sound

In this experiment, we also tested the influence of adding redundant discriminative features to letters on the final learning outcomes measured via letter discrimination, identification and words/pseudo-words reading. To this aim we compared two versions of the OVAL auditory orthography, tested in two different groups of participants who differed only in the presence/absence of color-specific information to OVAL letters, transformed into audition via timbre manipulations. Specifically, in the Monochromatic group, all letters were white, while in the Color group each letter was always appearing in a specific color (with three possible colors/timbres; Fig 1A).

While participants from both groups reached high accuracies in all experiments, performances of the Color group resulted systematically better. Specifically, participants from the Color group achieved success rates significantly higher than those achieved by participants in the Monochromatic group in all tasks. Reading speed was also significantly faster in the Color group, highlighting that accurate responses did not compromise reading speed. In certain tasks the color group achieved the ceiling effect, thus making the test insensitive to their superior performance compared to the Monochromatic group (Fig 2). Therefore, when using harder tasks, the advantage of the Color group over the Monochromatic group might even further increase. Facilitation of reading when adding redundant features has been shown for tactile pattern recognition, which appeared to be enhanced when texture was congruently added to shapes [52]. However, texture is difficult to systematically manipulate during tactile reading, and thus its contribution to reading was never directly assessed. The improvement in both reading speed and accuracy with the addition of redundant features to audemes, together with the ease of systematic addition of such features in the auditory modality compared with the tactile reading system, further supports the conclusion that auditory reading is a promising new venue. Future studies may further investigate the properties of such redundancy facilitation by investigating its limits. For instance, in this work we only used three colors, and color manipulations involved whole letters (Fig 1A). Will the facilitation be further enhanced by adding more colors/timbres? Or will it be further enhanced if we use timbre to differently sonify the most complex letters' elements to increase letters' discriminability–rather than full letters as was done here? In other words, which is the highest number of colors/timbres and which are the best letters' attributes to apply such manipulation, so that we would still observe a reading advantage before this redundant feature becomes detrimental for reading

performances? Future studies may further investigate this intriguing issue by aiming at further differentiating OVAL characters among each other through shape manipulations rather than only through the addition of redundant features such as color. In the current version of OVAL, for instance, letters were created with "descending" components, namely each component was positioned in a lower spatial position compared to the previous one, and thus were played with an increasing lower tone (Fig 1A). Future studies could, therefore, manipulate letters' shapes and create, for instance, not only "descending" characters as was the case here, but also "ascending" ones (i.e., characters which consecutive components are positioned in higher positions in space and thus played with an increasing higher tone). Additionally also more nuanced manipulations can be implemented such as changing the vertical distance between the components in letters that have a dual component (i.e., two components positioned on the same spatial column which are played together by the EyeMusic such as "L" or "P"; see Fig 1A). These studies, in turn, will allow to further uncover which manipulations (or combination of manipulations) will lead to the highest increase in OVAL characters' discriminability, ultimately also uncovering the threshold which if overcome, will lead to a decrement in reading performance.

Interestingly, our results showing enhancement of reading performance when color (i.e., timbre) was added to audemes are in contrast to some of the results obtained when adding color to visually presented letters. A recent study [68] explored the effect of reading with the addition of color in dyslexics as well as normal readers, both adults and children. When each whole-word was colored in a different color, reading was enhanced in all groups compared to a monochromatic condition, possibly due to the fact that color strengthened the perceptual grouping of words. However, when each letter was assigned a different color, similarly to our experiment, reading speed and error-rate returned to "baseline" levels, same as in monochromatic condition, for all 4 groups. The authors conclude that when coloring all letters, letters become "similarly dissimilar" just as in the monochromatic condition, so the reading enhancement achieved via the different coloring of each word, cancels out with the breaking of similarity achieved when each whole-word is colored in the same color [68].

This difference on the effects of redundant letters' features between visual and OVAL reading, may stem from the fact that individuals in the OVAL Color group, learned the letter already with the color assigned to them, thus binding the letter and the color. In contrast, in vision, individuals were introduced to the coloring of letters after many years of fluent reading. Another possibility is that the OVAL results may show that the addition of redundant characters' features might facilitate letters discriminability in the initial stages of learning to read with a novel orthography. Thus, it could be that after more reading training, redundant features will diminish their impact on OVAL reading outcomes. A third option is that unlike in vision, the addition of redundant features to an auditory spatio-temporal pattern, does not break the similarity principle, due to the serial nature of the OVAL presentation, opposed to the parallel presentation of print words [69]. However, due to the few studies done in vision to investigate the effect of adding redundant features such as color to letters, it is still too early to conclusively compare the outcomes of such addition on visual versus auditory reading. One interesting study to address this issue might be to test color-grapheme synesthets and ask whether they would also show better reading performances similarly to OVAL readers, at least during reading acquisition.

In addition, as expected, our results demonstrate "word superiority effect", namely words were read faster and more accurately than pseudo-words, in both Monochromatic and Color OVAL groups, even though accuracy in pseudo-words reading dropped more in the Monochromatic than in the Color group (Fig 3B). This is in line with previous results reporting faster and more accurate outcomes for words than pseudo-words also for visual and Braille

reading [70]. Interestingly, the length effect of Braille reading, in which the time to correctly read a word is significantly longer with every syllable added [70], was only evidenced in Monochromatic OVAL readers and not in Color OVAL readers. In other words, colors readers did not take significantly longer to read long than shorts OVAL strings, while Monochromatic readers took longer to read long strings (Fig 3E and 3F). This may further confirm the discriminative advantage of Color OVAL which seems to allow quicker long-strings processing. While the small pool of words as well as the short training period of participants alongside the relatively small sample size all limit the extent of application of the results, these lastly presented results suggest that the OVAL system may be a promising tool to investigate reading processes between sensory-modalities, ultimately allowing to disentangle sensory-specific from sensory-independent processes, an issue often debated in reading research for various reading components [71]. Past attempts in this direction compared performances between visual reading and Braille [71–73]. However, all this research had an inherent bias, as sighted individuals were tested on print while blind readers on Braille. Due to the inherent impossibility to test blind on print reading and to the complexity of learning Braille for sighted, none of the subjects could be tested on the other modality. The ease of learning auditory OVAL opens the intriguing possibility of directly comparing reading performances of sighted individuals in two modalities– print and audition, via OVAL reading. Future studies could therefore continue to assess the efficacy of OVAL as a reading system including also direct comparisons between OVAL and other visual print systems.

## Can OVAL be used to assist visual reading?

The potential of an auditory orthography for reading goes beyond blindness and includes sighted populations, especially people with specific reading impairments, such as dyslexic individuals. Indeed, the ultimate impairment leading to developmental dyslexia is still debated [74], but one of the most prominent views attributes dyslexia to an impairment in the cross-modal visual-auditory integration between graphemes and phonemes [75]. However, there is still no agreement on a gold-standard rehabilitation procedure for this reading impairment. Current rehabilitation programs for dyslexia taps on different skills spanning from potentiation of the auditory-phonological processing [76, 77], explicit systematic training on letter-to-speech correspondences [76, 78, 79], to training on visuo-attentional abilities via action videogames [80] or virtual reality [81]. These latter approaches are lately considered among the most promising venues, as it has been shown that individuals with developmental dyslexia suffer from specific visuo-spatial attention impairments which are evidenced even before children learn to read [82], the identification of which can potentially allow early interventions. Crucially, it has been proposed that these impairments cause the documented higher susceptibility to crowding effects of letters and words in dyslexics [83], ultimately preventing the deployment of the correct cross-modal grapheme-to-phoneme integration.

We suggest here that OVAL might aid dyslexic individuals in reading, perhaps even facilitating their learning of a reading system, by by-passing all the issues related to visual-attention documented in dyslexic individuals [82–84]—although some results seem to suggest that also auditory and not only visual attention might be impaired in these individuals [85, 86].

One option to test these conclusions might be to train dyslexic individuals in reading via audition using the OVAL, ultimately succeeding to separate the visual modality as reading channel, from the phonological system, thus directly testing the actual role of vision in reading impairments.

In addition, OVAL could be inserted within multisensory rehabilitative procedures pairing together both auditory and visual reading. As a matter of fact, dyslexics were shown to have

deficits in multisensory integration [86], however results suggest that they show sluggish shifts in cross-modal attention from vision to audition but not from audition to vision [86]. Thus, multisensory training with the OVAL might be undertaken in an interleaved rather than in a simultaneous fashion: in other words, dyslexics patients should be first exposed to OVAL sentences and then to the same text presented visually (see also [86] for a similar suggestion). The success of this training regimen should be tested on the improvement at the end of the program in print reading alone. Future studies may test the feasibility of this approach and perhaps adapt the OVAL algorithm to the specific needs of this population, for instance, by enlarging the spaces between letters [83] and by reducing the speed of letter presentation to match it more closely to the speed of reading in the visual modality, which is known to be slackened in this population [87]. In addition, these studies should also address text comprehension abilities on top of single-word reading speed. Importantly, also other groups could benefit from multisensory OVAL and visual reading, such as people with low vision, who currently can only read via pairing vision and Braille with all the issues highlighted above regarding the learning of Braille code, as well as people with progressive visual loss disorders such as patients with Retinitis Pigmentosa. These individuals could be trained on visual and auditory OVAL in parallel, perhaps making the OVAL training even quicker. Future studies may investigate these new venues and also test the extent to which pairing the print system with the OVAL system may slow down the process of visual deterioration.

## Concluding remarks

Taken together, our results suggest that due to its ease of learning and simple application, the auditory script OVAL, and especially its multi-color version, is an alternative to those who cannot read using the visual modality such as blind and low-vision individuals and could potentially be a rehabilitative tool for individuals with reading disabilities. We are currently working on the development of an OVAL app, which will offer the possibility to adapt OVAL reading features to the specific needs of different populations/users by allowing for instance, the personalized control and manipulation of the time of scanning to match reading speed of each individual. This includes both the speed of presentation of each letter (that can be even doubled) as well as the adjustment of spacing between letters and between words, among other features. In addition, OVAL script could be used to test the extent to which reading properties are sensory-independent or sensory-specific, due to its relatively easy learning which makes it potentially suitable for developmental longitudinal studies as well. OVAL is also applicable in the research of the reading network independent of visual exposure, and for research aiming to uncover potential differences due to serial versus holistic reading.

## Author Contributions

**Conceptualization:** Roni Arbel, Amir Amedi.

**Data curation:** Roni Arbel, Benedetta Heimler.

**Formal analysis:** Roni Arbel, Benedetta Heimler.

**Funding acquisition:** Amir Amedi.

**Investigation:** Roni Arbel, Benedetta Heimler.

**Methodology:** Roni Arbel, Benedetta Heimler.

**Project administration:** Roni Arbel.

**Supervision:** Benedetta Heimler, Amir Amedi.

**Writing – original draft:** Roni Arbel, Benedetta Heimler, Amir Amedi.

**Writing – review & editing:** Roni Arbel, Benedetta Heimler, Amir Amedi.

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
