## [Decision Letter · Decision Letter 0]

24 Jul 2020

PONE-D-20-15823

The Sound of Reading: Color-to-timbre substitution boosts reading performance via OVAL, a novel auditory orthography optimized for visual-to-auditory mapping.

PLOS ONE

Dear Dr. Arbel,

Thank you for submitting your manuscript to PLOS ONE. After careful consideration, we feel that it has merit but does not fully meet PLOS ONE’s publication criteria as it currently stands. Therefore, we invite you to submit a revised version of the manuscript that addresses the points raised during the review process.

The reviewers listed a number of specific points to be amended. In particular, the small sample size (N=18) is better to be justified (perhaps by performing a power-analysis) and the descriptions of statistics should be rechecked for the reviewers to properly evaluate a revised paper. Please provide point-by-point responses to the reviewers' comments. 

We look forward to receiving your revised manuscript.

Kind regards,

Katsumi Watanabe

Academic Editor

PLOS ONE

Journal Requirements:

2. We note that Figure1  in your submission contain copyrighted images. All PLOS content is published under the Creative Commons Attribution License (CC BY 4.0), which means that the manuscript, images, and Supporting Information files will be freely available online, and any third party is permitted to access, download, copy, distribute, and use these materials in any way, even commercially, with proper attribution. For more information, see our copyright guidelines: http://journals.plos.org/plosone/s/licenses-and-copyright.

2.1.         You may seek permission from the original copyright holder of Figure 1 to publish the content specifically under the CC BY 4.0 license.

2.2.    If you are unable to obtain permission from the original copyright holder to publish these figures under the CC BY 4.0 license or if the copyright holder’s requirements are incompatible with the CC BY 4.0 license, please either i) remove the figure or ii) supply a replacement figure that complies with the CC BY 4.0 license. Please check copyright information on all replacement figures and update the figure caption with source information. If applicable, please specify in the figure caption text when a figure is similar but not identical to the original image and is therefore for illustrative purposes only.

Reviewers' comments:

Reviewer's Responses to Questions

**Comments to the Author**

1. Is the manuscript technically sound, and do the data support the conclusions?

Reviewer #1: Yes

Reviewer #2: Yes

2. Has the statistical analysis been performed appropriately and rigorously? 

Reviewer #1: Yes

Reviewer #2: Yes

3. Have the authors made all data underlying the findings in their manuscript fully available?

Reviewer #1: Yes

Reviewer #2: Yes

4. Is the manuscript presented in an intelligible fashion and written in standard English?

Reviewer #1: Yes

Reviewer #2: Yes

5. Review Comments to the Author

Reviewer #1: I have answered 'yes' in good-faith to question 3, due to the fact that the article says that they will make the data available upon acceptance of the manuscript, as such the data was not available during the time of the review and I was only provided with the manuscript pdf.

Reviewer #2: The manuscript introduces OVAL, a character-based visual-to-audio sensory substitution algorithm based on the existing eyeMusic application. OVAL is a character system, or auditory character based reading system, that uses the eyeMusic for processing the ‘audemes’. The authors posit this as a tool to help non-literacy in non-visual readers, in response to the dwindling numbers of Braille readers. The authors test the efficiency of OVAL with 18 sighted participants after training for 6 hours. The participants were split into two groups, monochromatic and colour, to see whether a greater range of timbres could help further improve auditory reading. The results suggest that colour information that is subsituted into timbre can help add 'texture' and improve reading accuracy and speed. The paper suggests that OVAL can be learnt faster than Braille and may assist in the learning of non-visual reading.

I liked the paper a lot and think it makes a valuable contribution to the field. I particularly like the demonstration of colour-to-timbre substitution for improving the speed and accuracy of performance, as colour substitution is generally under-represented (or even written off!) in SSD research. More work into using colour as an extra layer of distinguishing multiple features apart will be really useful for the field.

I do think the narrative needs to be tweaked slightly. At the moment, there feels like there is a large focus is on the practical uses of OVAL, in that, it has the potential to be more useful/available/easier to learn than Braille, and the authors make a compelling case for as such. However, with such a focus on the practical aspects of OVAL, there needs to be an earlier and thorough address of how OVAL stands apart from screen readers and audiobooks. It is only late in the discussion that screen readers or audiobooks gets mentioned, but I spent the whole paper thinking about them.

The authors mention how Braille is dying out, but I would possibly say that this could be related to the increased availability of audio technology. I know that reading through OVAL is a different skill, and will offer different cognitive development to listening to spoken word, but this should be made very clear. I would like to see some specific comparisons of past literature about the general benefits of reading (for knowledge, information etc.) against the benefits of specifically reading through characters (spelling, grammar etc.). Basically, I want to know in more detail why the declining Braille literacy is a problem, and why screen readers and audiobooks can’t fully solve this problem. Is Braille dying out simply because audiobooks and screen readers are easier and quicker to attain? This would do well in the introduction, so that your readers are immediately on board with why OVAL is important, and why blind and visually impaired readers should use it at the cost of slower text acquisition.

Also, the authors need to say why audemes are better than just playing audio recordings of spoken individual letters. Each letter has its own sound and would represent an ‘audeme’ that is already familiar. Is OVAL just like learning a purely auditory character language?

I suspect the PLOS ONE will address formatting issues through publication, but if you need to use the accepted version for anything, I would recommend checking titling and heading formats in the paper as there are some inconsistencies. For a specific example, in the methods section, behavioural experiments (page 16), ‘Experiment 1 – letter discrimination’ has no capitalisation on the ‘l’ but ‘Experment 2 – Discrimination of trained vs. untrained letters’ has capitalisation on the ‘d’. Also, there’s the typo on ‘experiment 2’. Then later on in the manuscript, in the results section (page 18), ‘Experiment 1 - Letter discrimination task’ is not in italics which doesn’t pair with the methods section to well, and the ‘l’ is now capitalised.

I would also check formatting with reporting statistics in text. For example, on page 19, the manuscript has a mixture of spacing between statistic, operator, and number: ‘([F(1,16) = 11.3, p = 0.004])’ and ‘=11.70, p<0.00001; Color readers t=158.55, p<0.00001’. I would also check general conventions for reporting statistics. On PLOS ONE’s guideline page they have some useful info. Like reporting p-values less than 0.001 as p < 0.001 rather that trailing zeros, and stating whether one or two tailed test were used. It’s really handy for readers to be reminded of effect sizes too. On page 20, I assume you’re using cohen’s d, remind the audience what is considered a big/medium effect size. Also, double check which statistics should be in italics. Here’s the guide as a starting point: https://journals.plos.org/plosone/s/submission-guidelines.#loc-statistical-reporting

Those are the main points that need addressing, and here are some specifics:

I think having the first line of the introduction being the same as the first line of the abstract damages readability a little bit, consider rewording?

On page 11, in the participants section, I think it’s useful to briefly justify why sighted participants rather than blind. Other researchers in the area will understand why, but those reading from outside may question why a Braille-like system is being tested with sighted.

Page 14, 2 hours for each training session seems like a long time, maybe add a note on participants fatigue or if there was a break.

Page 17, I don’t think you need to repeat ‘i.e., the time used by the EyeMusic to scan each OVAL display’. After the first time, it was clear that four second per trial was due to the speed of the scan.

Page 18, state which JASP version you used if available.

Page 19, ‘This suggests that even without any phonological association to the auditory patterns, basic auditory discrimination of the soundscapes was greatly enhanced by the addition of colorconveyed via timbre manipulations(see also (32)for another color-related advantage in soundscapes discriminations)’ – I would get rid of this from the results and move the discussion. This sounds like interpretation of the results, rather than raw results.

Page 20, there’s a disparity between the reaction time and character repetition. The manuscript states that both groups required one character repetition, but their response times were both under 4 second. Do you mean one character presentation rather than repetition? The character was presented one time only rather than repeated. This would make sense with the reaction times, otherwise I would expect the reaction times to be over 4 seconds.

Page 22, ‘Finally, and most importantly, also the interaction between…’. Again, I feel like the word 'importantly' is adding some interpretation to the raw results. There hasn’t been any justification for this being more important than the other results.

Discussion – I think the authors need to tone down some of the claims slightly. Namely, that the results demonstrate that audio reading can be ‘mastered’ in adulthood. It’s too earlier to say whether your participants mastered OVAL reading; a longer, more in depth training procedure would be needed to see possible performance. Actually, this claim does a disservice to the study, as with a lot more practice your participants may get much better! The discussion also goes into detail about accuracy in Braille reading studies, but not speed of reading. Previously, the manuscript mentions that the colour OVAL does not suffer with a speed accuracy trade off (so speed of reading is important), but then neglects speed comparisons with Braille.

The declining Braille section (page 25) should potentially be in the introduction. While the intro does mention that Braille is declining it doesn’t give these specifics which would help justify OVAL slightly more thoroughly. Also, later on the page, the sentence ‘Several previous studies conducted on blindfolded sighted Braille learners, provide convergent results supporting these conclusions’ needs references at the end of the sentence.

Page 27, I really like this section. It would be cool to see any comparisons to synesthetes reading if there’s any applicable research! Do colour-grapheme synesthetes have increased comprehension because of colour information, like with OVAL? Not sure if this has been done to be honest.

Page 30, the authors need to address reading comprehension here. Slow reading (even in dyslexics) reduces comprehension, and the relative slowness of reading by OVAL (compared to slighted reading) may actually hinder reading comprehension.

Figures are great! Just check capitalisation again, some variables start with capital letters and other don’t.

6. PLOS authors have the option to publish the peer review history of their article (what does this mean?). If published, this will include your full peer review and any attached files.

Reviewer #1: **Yes: **Giles Hamilton-Fletcher

Reviewer #2: **Yes: **Mike Richardson

---

## [Author Response · Author response to Decision Letter 0]

12 Oct 2020

1. We added a power analysis in relation to the sample size.

2. We provide a link for where the data can be found

3. Figure 1 on the manuscript, including all its elements, was developed specifically for this manuscript by the authors, its copyrights are held by the lab.

4. We addressed all comments and requests by the editor and reviewers in the file "response to reviewers" and revised the manuscript accordingly. 

5. We formatted the revised manuscript to meet PlosOne style requirements.

---

## [Decision Letter · Decision Letter 1]

6 Nov 2020

The sound of reading: Color-to-timbre substitution boosts reading performance via OVAL, a novel auditory orthography optimized for visual-to-auditory mapping.

PONE-D-20-15823R1

Dear Dr. Arbel,

We’re pleased to inform you that your manuscript has been judged scientifically suitable for publication and will be formally accepted for publication once it meets all outstanding technical requirements.

Kind regards,

Katsumi Watanabe

Academic Editor

PLOS ONE

Additional Editor Comments (optional):

There are a few minor comments by the reviewers that may amend the manuscript further. The authors may want to consider these comments before providing the final version.

Reviewers' comments:

Reviewer's Responses to Questions

**Comments to the Author**

1. If the authors have adequately addressed your comments raised in a previous round of review and you feel that this manuscript is now acceptable for publication, you may indicate that here to bypass the “Comments to the Author” section, enter your conflict of interest statement in the “Confidential to Editor” section, and submit your "Accept" recommendation.

Reviewer #1: All comments have been addressed

Reviewer #2: All comments have been addressed

2. Is the manuscript technically sound, and do the data support the conclusions?

Reviewer #1: Yes

Reviewer #2: Yes

3. Has the statistical analysis been performed appropriately and rigorously? 

Reviewer #1: Yes

Reviewer #2: Yes

4. Have the authors made all data underlying the findings in their manuscript fully available?

Reviewer #1: Yes

Reviewer #2: Yes

5. Is the manuscript presented in an intelligible fashion and written in standard English?

Reviewer #1: Yes

Reviewer #2: Yes

6. Review Comments to the Author

Reviewer #1: The authors have thoroughly and robustly addressed the points raised by my peer review and I consider the manuscript in a sound state for publication. Please see the reviewer attachment for my responses to each individual point.

Reviewer #2: The authors have addressed all my previous comments in a agreeable way, and the revised version of the manuscript is great.

I thank the authors for their effort in revising the manuscript, and making an interesting contribution to the field.

7. PLOS authors have the option to publish the peer review history of their article (what does this mean?). If published, this will include your full peer review and any attached files.

Reviewer #1: **Yes: **Giles Hamilton-Fletcher

Reviewer #2: **Yes: **Mike Richardson

---

## [Editor Report · Acceptance letter]

13 Nov 2020

PONE-D-20-15823R1 

The sound of reading: Color-to-timbre substitution boosts reading performance via OVAL, a novel auditory orthography optimized for visual-to-auditory mapping. 

Dear Dr. Arbel:

I'm pleased to inform you that your manuscript has been deemed suitable for publication in PLOS ONE. Congratulations! Your manuscript is now with our production department. 

Kind regards, 

on behalf of

Dr. Katsumi Watanabe 

Academic Editor

PLOS ONE